# Delivering Two Tumour Antigens Survivin and Mucin-1 on Virus-Like Particles Enhances Anti-Tumour Immune Responses

**DOI:** 10.3390/vaccines9050463

**Published:** 2021-05-06

**Authors:** Katrin Campbell, Vivienne L. Young, Braeden C. Donaldson, Matthew J. Woodall, Nicholas J. Shields, Greg F. Walker, Vernon K. Ward, Sarah L. Young

**Affiliations:** 1Department of Pathology, Otago Medical School, University of Otago, Dunedin 9016, New Zealand; katrin.campbell@otago.ac.nz (K.C.); Braeden.donaldson@otago.ac.nz (B.C.D.); matthew.woodall@otago.ac.nz (M.J.W.); 2Department of Microbiology and Immunology, School of Biomedical Sciences, University of Otago, Dunedin 9016, New Zealand; vivienne.young@otago.ac.nz (V.L.Y.); vernon.ward@otago.ac.nz (V.K.W.); 3Faculty of Medicine and Health, School of Medical Sciences, The University of Sydney, Sydney 2006, Australia; nicholas.shields@sydney.edu.au; 4School of Pharmacy, University of Otago, Dunedin 9016, New Zealand; greg.walker@otago.ac.nz

**Keywords:** breast cancer, nanoparticles, virus-like particles (VLPs), conjugated VLPs, multimeric presentation, therapeutic vaccines, cancer immunotherapy

## Abstract

Breast cancer (BC) is the most frequently diagnosed cancer in women, with many patients experiencing recurrence following treatment. Antigens delivered on virus-like particles (VLPs) induce a targeted immune response and here we investigated whether the co-delivery of multiple antigens could induce a superior anti-cancer response for BC immunotherapy. VLPs were designed to recombinantly express murine survivin and conjugated with an aberrantly glycosylated mucin-1 (MUC1) peptide using an intracellular cleavable bis-arylhydrazone linker. Western blotting, electron microscopy and UV absorption confirmed survivin-VLP expression and MUC1 conjugation. To assess the therapeutic efficacy of VLPs, orthotopic BC tumours were established by injecting C57mg.MUC1 cells into the mammary fat pad of mice, which were then vaccinated with surv.VLP-SS-MUC1 or VLP controls. While wild-type mice vaccinated with surv.VLP-SS-MUC1 showed enhanced survival compared to VLPs delivering either antigen alone, MUC1 transgenic mice vaccinated with surv.VLP-SS-MUC1 showed no enhanced survival compared to controls. Hence, while co-delivery of two tumour antigens on VLPs can induce a superior anti-tumour immune response compared to the delivery of single antigens, additional strategies must be employed to break tolerance when targeted tumour antigens are expressed as endogenous self-proteins. Using VLPs for the delivery of multiple antigens represents a promising approach to improving BC immunotherapy, and has the potential to be an integral part of combination therapy in the future.

## 1. Introduction

Immunotherapy is a fast-evolving field that has gained increased attention in recent years due to its clinical successes, especially that of immune checkpoint blockade. Breast cancer (BC) has been a target for immunotherapy for decades—from the clinical success of treating HER2 + BC with the monoclonal antibody Trastuzumab [1] to the approval of the PD-L1 inhibitor atezolizumab for triple-negative BC (TNBC) in 2019 [2]. Although great progress has been made in immunotherapeutic research, treatments currently available are not effective in all patients.

A promising approach to specifically target BC cells while sparing healthy cells is the active induction of an anti-cancer immune response using peptide vaccines. Induction of a strong and reliable anti-tumour immune response can be achieved by delivering tumour antigens as well as vaccine adjuvants to the same antigen-presenting cell (APC) [3]. Using this strategy, APCs are strongly activated by adjuvant and upregulate co-stimulatory molecules, while the co-delivered antigens are processed and loaded onto major-histocompatibility complex (MHC) molecules for presentation to cytotoxic T lymphocytes (CTL) and T helper (Th) cells [4]. The co-delivery of two antigens to form a multi-target vaccine decreases the risk of tumour escape by targeting a wider range of tumour cell populations, and offers a potential treatment avenue for BCs given their heterogeneity [5]. This concurrent delivery of peptides can be achieved by using nanoparticle vehicles, such as virus-like particles (VLPs). VLPs have been proven safe as vaccine formulations, as they act both as a vaccine platform to deliver tumour antigens and induce potent cellular immune responses by directly interacting with APCs [6,7,8]. As a delivery platform, VLPs are amenable to a broad range of modifications, including recombinant insertion, chemical conjugation, encapsulation, and surface adsorption [9].

Several tumour-associated antigens (TAAs) have been identified as promising targets for cancer immunotherapy, including mucin-1 (MUC1) [10] and survivin [11,12,13]. MUC1 is a transmembrane glycoprotein that is overexpressed in approximately 90% of BCs. Notably, compared to MUC1 expressed by healthy cells, cancer-derived MUC1 is aberrantly glycosylated, exhibiting GalNAcα1-Ser/Thr modification (Tn antigen) and other truncated O-glycans due to a lack of core 1 β3-galactosyltransferase (T-synthase) activity [14,15,16]. The aberrant glycosylation of MUC1 in both cancers and premalignant lesions makes it a valuable target, which was recognised in a National Cancer Institute (NCI) report from 2009 ranking MUC1 as one of the most promising peptides for cancer immunotherapy [17,18]. Survivin is another TAA that is overexpressed in up to 90% of breast cancers [19]. Survivin has been previously investigated as an immunotherapeutic target in a range of tumour types. However, there are limited reports on its use as a target in BC immunotherapy [13,19,20]. Furthermore, while MUC1 and survivin peptides have been investigated as immunotherapeutic targets individually, the potential synergistic effect of co-delivering them in a multi-target BC vaccine has not been examined.

In this study, we developed a VLP-based BC vaccine that co-delivered both the MUC1 and survivin antigens, as well as CpG as a vaccine adjuvant. VLPs derived from rabbit haemorrhagic disease virus (RHDV) are composed of 180 copies of the viral capsid protein, VP60, and each VP60 unit can be recombinantly engineered to express tumour epitopes, such as survivin_97-104_ (TVSEFLKL), to produce surv.VLP [21]. In addition to the incorporation of survivin peptide via recombinant expression, we used a previously reported intracellular reversible conjugation strategy to chemically conjugate the aberrantly glycosylated MUC1 peptide, SAPDT(GalNAc)RPAPGST(GalNAc)APPA, to the surface of RHDV VLPs expressing survivin_97-104_ [3]. The resultant surv.VLP-SS-MUC1 particle was subsequently tested in an orthotopic murine BC model to evaluate the therapeutic effect of simultaneously targeting two tumour antigens.

## 2. Materials and Methods

### 2.1. Generation of RHDV VLP

RHDV VLP (VP60) and RHDV VLP engineered to express the tumour antigen epitope survivin_97-104_ (Surv.VLP) were developed using a recombinant baculovirus expression system, as previously described [21,22]. In brief, the DNA sequence encoding the survivin peptide was purchased as synthetic DNA in a pUC57-Simple plasmid (Genscript, Piscataway Township, NJ, USA). The survivin sequence was extracted and ligated with the RHDV VP60 in a pAcUW51 (GUS) expression plasmid. The expression plasmids (VP60 and Surv.VLP) were co-transfected with the FlashBAC ULTRA™ (Oxford Expression Systems, Oxford, UK) into *Spodoptera frugiperda* (Sf21) cells. To express VP60 and Surv.VLP, suspension cultures of Sf21 cells were infected with the respective recombinant baculovirus at a multiplicity of infection of 1 and incubated for 3 days at 27 °C with shaking at 125 rpm. To isolate VLPs, the cells were lysed with 0.5% Triton X-100, followed by differential centrifugation and a CsCl gradient using ultracentrifugation (100,000× *g*, 18 h, 4 °C). The VLP band was harvested and the VLP concentration determined by A280 absorbance by NanoDrop (Thermo Scientific, Rockford, IL, USA) using an extinction coefficient of 78,000 M^−1^ cm^−1^ and a molecular weight of 60,000 Da.

### 2.2. Confirmation of VLP Expression

To confirm VP60 expression, VLPs were separated by sodium dodecyl sulphate-polyacrylamide gel electrophoresis (SDS-PAGE) followed by transfer onto a nitrocellulose membrane using a Transblot SD blotter (Bio-Rad, Hercules, CA, USA). The membrane was blocked with 0.1% casein alanate for 1 h, probed by Western blotting with rabbit anti-VP60 (University of Otago, Dunedin, New Zealand) and DyLight 800-labelled donkey anti-rabbit monoclonal antibodies (Clone SA5–10044, Lot QC1998302, Thermo Scientific, Wilmington, DE, USA) and then imaged on an Odyssey FC (Licor, St, Lincoln, NE, USA). To confirm self-assembly of VP60 and Surv.VLP into particles, VLPs were negatively stained with phosphotungstic acid and visualized using a Philips CM100 transmission electron microscope at the Otago Centre for Electron Microscopy.

### 2.3. Mass Spectrometry

To confirm survivin recombination, surv.VLP were run on a 10% SDS-PAGE gel and visualized by Coomassie Brilliant Blue G-250 staining (BD Biosciences, San Jose, CA, USA). The resulting band was excised and the survivin epitope was identified by mass spectrometry using matrix-assisted laser desorption/ionization time of flight (MALDI-TOF/TOF, Centre for Protein Research, University of Otago, Dunedin, New Zealand).

### 2.4. Conjugation of MUC1 Peptide to VLP

Purified VLPs (VP60 and surv.VLP) were dialysed (14 kDa MWCO cellulose membrane, Sigma, St Louis, MO, USA) against coupling phosphate buffered saline cPBS1 (0.1 M sodium phosphate and 0.15 M NaCl, pH 7.4). For conjugation, the VLPs were modified with an 8-fold molar excess of succinimidyl-SS-4-formyl- benzoate (S-SS-4FB, TriLink Biotechnologies, San Diego, CA, USA) in cPBS1 for 1.5 h at room temperature on a rotor. The unreacted linker was removed by overnight dialysis against cPBS2 (0.03 M NaH_2_PO_4_, 0.17 M Na_2_HPO_4_ and 0.15 M NaCl, pH 6.5). MUC1 peptide (SAPDT(GalNAc) RPAPGST(GalNAc)APPA) was modified with a 0.9-fold molar excess of succinimidyl 6-hydrazinonicotinate acetone hydrazone (S-HyNic from TriLink) in cPBS1 for 1.5 h at room temperature on a rotor. The S-SS-4FB-modified VLP was then reacted with the S-HyNic-modified MUC1 peptide in cPBS2 including 10 μM aniline for 1.5 h at room temperature on a rotor to form a stable bis-arylhydrazone bond. Unreacted peptide and linker were removed by overnight dialysis against cPBS pH 7.4. Dialysed VLPs were stored in 50% glycerol, 50% cPBS1 at −20 °C. To confirm MUC1 conjugation, the VLP was diluted in cPBS1 and analysed by measuring the UV–Vis absorbance at 354 nm to detect the formation of the bis-arylhydrazone bond.

### 2.5. Conjugate Cleavage Assay

Glutathione (GSH) was prepared in PBS at a stock concentration of 50 mM. surv.VLP-SS-MUC1 were diluted using cPBS1 and GSH stock to prepare samples at 2 mg/mL in either cPBS1 or 5 mM GSH. The samples were incubated for 2 h at 37 °C and then analysed by measuring the UV–Vis absorbance at 354 nm to determine the conjugation ratio.

### 2.6. Animals: Source and Ethics

Specific pathogen-free C57BL/6 mice were sourced from the Hercus Taieri Research Unit, University of Otago, Dunedin, New Zealand. A colony of MUC1.Tg mice (C57BL/6-Tg(MUC1)79.24Gend/J, The Jackson Laboratory, Bar Harbour, ME, USA) was maintained by crossing MUC1.Tg mice with C57BL/6 mice. Mice were genotyped by standard polymerase chain reaction (PCR) using DNA isolated from ear notches with the following primers: forward, 5′-CTTGCCAGCCATAGCACCAAG-3′; reverse, 5′-CTCCAC- GTCGTGGACATTGATG-3′. The DNA product of each reaction was analysed by size fractionation through a 1% agarose gel. The size of the DNA product from MUC1-positive mice corresponded with a 500 bp fragment [23]. MUC1 transgenic mice were maintained as hemizygous animals. Experiments were conducted in accordance with ethical permits granted by the University of Otago Animal Ethics Committee (AEC 14/17). All animals were euthanised by cervical dislocation or carbon dioxide euthanasia.

### 2.7. In Vivo Cytotoxicity

Groups of 6 female C57BL/6 mice aged 7–9 weeks were vaccinated with cPBS or 100 μg of VP60, surv.VLP, VP60-SS-MUC1 or surv.VLP-SS-MUC1, combined with 25 μg CpG 1826 (GeneWorks, SA, Australia). Each treatment was administered subcutaneously into the left flank in 100 μL solution. After 21 days, a boost of the same treatment was given to each mouse. Target cells were prepared from C57BL/6 donor mice splenocytes. Red blood cells were lysed with ammonium chloride buffer, and the remaining white blood cells were separated into four populations. The cell populations were either left unpulsed or pulsed with 10 μM of either SV (survivin antigen), SAP (MUC1 antigen) or both SV and SAP. Following 2 h incubation at 37 °C, 5% CO_2_, the populations were stained with one of the following dye combinations: 5 μM carboxyfluorescein succinimidyl ester (CFSE^Lo^), 50 μM CFSE (CFSE^Hi^), 4 μM violet proliferation dye (VPD) or 50 μM CFSE + 4 μM VPD (CFSE^Hi^/VPD). Target cells were injected into mice intravenously. After 40 h, spleens from vaccinated mice were harvested, the red blood cells lysed with ammonium chloride buffer and the remaining lymphocytes stained with LIVE/DEAD Fixable Near-IR Dead Cell Stain (Invitrogen, Carlsbad, CA, USA). Fluorescence was measured using a Gallios flow cytometer (Beckman Coulter, Brea, CA, USA) with a three-laser (405 nm, 488 nm, and 633 nm), ten-colour configuration and analysed using Kaluza software (Beckman Coulter, Brea, CA, USA). Percentage killing was calculated with the following formula:$$\%\;specific\;lysis\;=\;\left[1\;-\;\frac{\left(\frac{target\;cell\;\#}{control\;cell\;\#}\right)vaccinated}{\left(\frac{target\;cell\;\#}{control\;cell\;\#}\right)PBS}\right]\;*\;100$$

### 2.8. Tumour Cell Culture and Implantation

C57mg.MUC1 and C57mg.WT cells were kindly provided by Prof. Sandra Gendler, Mayo Clinic, Arizona, USA. C57mg.MUC1 cells were cultured in DMEM media (HyClone Laboratories, Logan, UT, USA), 1% GlutaMAX (Gibco, Rockville, MD, USA), 1% Pen/Strep (Gibco), 10% FCS (Moregate Biotech, Hamilton, NZ) and 150 μg/mL G418 (Gibco). C57mg.WT cells were cultured in DMEM, 1% GlutaMAX, 1% Pen/Strep and 10% FCS. Cells were split twice, and grown to 80% confluence each time before injection into mice. Cells were detached from culture flasks with 1% EDTA, washed three times in DPBS, and adjusted to the desired cell concentration in DPBS. Cells were kept on ice until injection. Mice were anaesthetised with ketamine/domitor/atropine and the fur in the upper left quarter of the abdomen removed by clippers and Nair depilatory cream (Church and Dwight, Princeton, NJ, USA). The second thoracic mammary fat pad was grasped with tweezers before 20 μL of the cell suspension was injected using a 31G insulin syringe. The mice were injected with antisedan for domitor reversal and monitored until they regained consciousness.

### 2.9. Tumour Cell Titration

C57BL/6 mice were injected with 0.5 × 10^5^, 1 × 10^5^, 5 × 10^5^ and 10 × 10^5^ C57mg.MUC1 or C57mg.WT cells and MUC1.Tg mice were injected with 0.5 × 10^5^, 1 × 10^5^, 5 × 10^5^ and 10 × 10^5^ C57mg.MUC1 cells. Mouse weight and tumour size were monitored every one to two days and mice were euthanised once tumours reached 150 mm^2^.

### 2.10. Therapeutic Tumour Trial

C57mg.MUC1 (0.5 × 10^5^) and C57mg.WT (1 × 10^5^) cells were injected into C57BL/6 mice (*n* = 10/group), and C57mg.MUC1 (0.5 × 10^5^) cells were injected into MUC1.Tg mice (*n* = 10/group) as described above. Once tumours were palpable on day 7, C57BL/6 mice were injected with either DPBS or 100 μg VLP (VP60, surv.VLP, VP60-SS-MUC1 or surv.VLP-SS-MUC1) with 25 μg CpG and MUC1.Tg mice were injected with either 100 μg VP60 or surv.VLP-SS-MUC1 with 25 μg CpG. Mouse weight and tumour size were measured every one to two days and mice were euthanised once tumours reached 150 mm^2^.

### 2.11. Tumour Infiltrating Lymphocytes

End-point tumours (150 mm^2^) from VP60 and surv.VLP-SS-MUC1 treated mice were resected for tumour infiltrating lymphocyte (TIL) analysis. Five tumours per group were fixed in formalin for CD3 immunohistochemistry (Histology Unit, University of Otago). Images were taken with Aperio CS2 and analysed using ImageJ software (National Institutes of Health). The other five tumours per group were processed into single-cell suspensions by cutting them into small pieces using a scalpel, then sequentially passing through 100 and 70 μm cell strainers, followed by density gradient centrifugation (Ficoll-Paque PLUS density gradient media, GE Healthcare). TILs were stained with Zombie-Yellow Live/Dead Stain, treated with CD16/CD32 Fc blocking antibody, and then stained with PD-1-FITC (clone 29F.1A12, BioLegend, San Diego, CA, USA), LAG3-PE (clone C9B7W, BD), CD127-PE/CF594 (clone SB/199, BioLegend), CD39-PE/Cy7 (clone Duha59, BioLegend), TIM3-APC (clone RMT3-23, BioLegend), CD8-AF700 (clone 53-6.7, BioLegend), CD4-APC/Cy7 (clone GK1.5, BioLegend) and CD3-BV421 (clone 17A2, BioLegend). Fluorescence was measured using a Gallios flow cytometer and analysed using Kaluza software (Beckman Coulter, Brea, CA, USA).

### 2.12. Statistical Analysis

Statistical analyses were performed using GraphPad Prism version 7.0 (GraphPad, San Diego, CA, USA). Statistical analysis was carried out using one-way analysis of variance (ANOVA) with Dunnett’s post hoc test to compare the difference of one variation in more than two different treatment groups, the Mann–Whitney test to compare the difference of one variation between two groups and a log-rank (Mantel–Cox) test to compare survival rates. The particular type of statistical analysis performed is indicated in each of the relevant figure legends. Error bars in graphs depict the standard error of the mean (SEM).

## 3. Results

### 3.1. VLP Design and Conjugation

The RHDV VP60 capsid protein was designed to express the survivin_97-104_ epitope TVSEFLKL based on published reports (surv.VLP, Figure 1a) [21,24]. Purification of both VP60 and surv.VLP were confirmed by SDS-PAGE gel and Western blot for the RHDV VP60 major capsid protein (Figure 1b). Expression of the survivin epitope in surv.VLP was confirmed by mass spectrometry (Appendix A). In order to allow delivery of two tumour-associated antigens concurrently, the MUC1 peptide SAPDT(GalNAc)RPAPGST(GalNAc)APPA was conjugated onto surv.VLP via an intracellular reducible linker. This peptide sequence with aberrant O-glycosylation on the threonines (Tn; N-Acetylgalactosamine threonine) was selected based on previously published reports showing immune responses towards aberrantly glycosylated MUC1 peptides from the VNTR region in mice transgenic for the human MUC1 [16,18,25,26,27,28].

For conjugate generation, the lysine residues on the VP60 capsid protein were modified with the aromatic aldehyde linker S-SS-4FB. While the MUC1 peptide was modified with the hydrazine linker S-HyNic. The aldehyde-modified surv.VLP was then reacted with the hydrazine-modified MUC1 peptide to generate a stable bis-aryl hydrazine bond (Figure 1d) [3]. The conjugation ratio of 3.8 VLP:MUC1 for surv.VLP-SS-MUC1 and VP60-SS-MUC1 was established by measuring the bis-aryl hydrazone bond formed by UV spectroscopy at 354 nm.

Cleavage of the intracellular reducible bis-aryl hydrazine bond in intracellular conditions was determined by incubating the conjugated surv.VLP-SS-MUC1 in either PBS alone or PBS containing 5 mM GSH as published in our previous studies [3]. Conjugation ratios of 3.2 for surv.VLP-SS-MUC1 in PBS and 0.56 for surv.VLP-SS-MUC1 in 5 mM GSH were determined by UV–Vis at 354 nm, confirming the reducibility of the disulphide bond at intracellular concentrations of GSH, and thereby the release of the MUC1 peptide inside cells.

VLPs from RHDV VP60 capsid proteins spontaneously assemble into cog-like particles of approximately 40 nm in diameter. Assembly of both surv.VLP and conjugated surv.VLP-SS-MUC1 was confirmed by electron microscopy (Figure 1c). The surv.VLP-SS-MUC1 particles showed increased clustering compared to the unconjugated surv.VLP particles.

### 3.2. Cytotoxicity Assay

VLPs delivering survivin and MUC1 peptides were tested to determine their ability to induce target antigen-specific cytotoxicity. Mice were vaccinated twice with VLPs delivering survivin and/or MUC1 peptides (or empty VLP controls) using CpG as an adjuvant. On day 28, mice were challenged with target cells pulsed with the survivin epitope, the MUC1 epitope, and non-pulsed control cells (Figure 2a). Splenocytes from vaccinated mice were analysed 40 h later in order to determine antigen-specific cytotoxicity against target cells (Figure 2b). Figure 2b shows that vaccination with surv.VLP and surv.VLP-SS-MUC1 in wild-type C57BL/6 mice—as well as vaccination with surv.VLP-SS-MUC1 in MUC1.Tg mice—induced specific cytotoxicity against the targeted survivin peptide. No off-target cytotoxicity against VP60 or VP60-SS-MUC1 was observed. Similarly, C57BL/6 mice vaccinated with VP60-SS-MUC1 and surv.VLP-SS-MUC1 as well as MUC1.Tg mice vaccinated with surv.VLP-SS-MUC1 exhibited specific cytotoxicity against the targeted MUC1 peptide, while no off-target cytotoxicity against VP60 or surv.VLP was observed. The administration of peptides alone induced lower cytotoxicity against both survivin and MUC1 pulsed cells, confirming that VLP-based delivery of peptides enhances the generation of antigen-specific immune responses in vivo.

### 3.3. Therapeutic Tumour Trial in Wild-Type Mice

VLPs were subsequently tested in a therapeutic tumour trial to determine whether they could elicit protection against an established tumour in wild-type C57BL/6 mice. First, C57mg.MUC1 breast cancer cells were confirmed to express MUC1 in culture (Figure 3a, left), as well as PD-L1 following interferon-γ (IFN-γ) stimulation (Figure 3a, right) by flow cytometry. Increasing numbers of tumour cells were implanted into the mammary fat pad of mice and 0.5 × 10^5^ C75mg.MUC1 and 1 × 10^5^ C57mg.WT cells were determined as the optimal cell number for tumour inoculation (see Appendix A). Next, mice were implanted with either C75mg.MUC1 cells or C57mg.WT cells to assess the therapeutic activity of VLPs. Once tumours were palpable, mice were injected with VLPs delivering either survivin, MUC1, both, or no antigen (*n* = 10 mice/group). Mice bearing control C57mg.WT tumours showed no delay in tumour growth when injected with the surv.VLP, while there was a small delay in tumour growth when mice were vaccinated with the double-antigen VLP (surv.VLP-SS-MUC1) compared to the empty VLP controls (VP60; see Appendix A). Figure 3c,d show that in mice implanted with the MUC1 expressing C57mg.MUC1 cells, vaccination with VLPs expressing both survivin and MUC1 (surv.VLP-SS-MUC1) delayed tumour growth and increased overall survival compared to PBS and empty VP60 VLP controls, as well as vaccination with VLPs expressing survivin (surv.VLP). Vaccination with VLPs delivering only MUC1 (VP60-SS-MUC1) provided some protection compared to vehicle and empty VP60 controls. However, there was no significant difference in survival compared to vaccination with surv.VLP. Vaccination with VLPs delivering survivin only did not significantly delay tumour growth when compared to PBS and empty VP60 VLP controls.

End-point tumours (having reached 150 mm^2^) from mice vaccinated with either control VP60 VLP or surv.VLP-SS-MUC1 were excised and analysed by flow cytometry to examine the expression of T-cell exhausted markers (*n* = 5) or analysed by immunohistochemistry (IHC) for CD3+ to examine T-cell infiltration (*n* = 5). For flow cytometry, isolated cells were gated on either CD3+ CD4+ cells or CD3+ CD8+ cells and then the percentages of PD-1+, CD39+, LAG-3+, TIM-3+ cells were analysed. Figure 3e shows that while there was a decreased proportion of CD4+ PD-1+ LAG-3+ T cells in surv.VLP-SS-MUC1 vaccinated mice (compared to VP60 controls), there were no other differences in exhausted T-cell populations. When examining T-cell infiltration by IHC, a trend of increased CD3+ cell counts in the VLP-vaccinated group compared to the VP60 control group was observed (in the total tumour area as well as the intratumoural area). However, these differences were not statistically significant. When comparing the density of CD3+ cells infiltrating tumours, a significantly higher density of T cells could be seen in the peritumoural area of VP60 control mice compared to VLP-vaccinated mice. There were no differences in CD3+ cell density in the whole tumour or intratumoural area between groups (Figure 3f).

### 3.4. Therapeutic Tumour Trial in MUC1 Transgenic Mice

Following the evaluation of our VLP in wild-type mice, its efficacy was subsequently assessed in mice transgenic for the human MUC1 protein [23]. In this model, C57mg.MUC1 cells were implanted into transgenic MUC1 mice (MUC1.Tg). Mice vaccinated with surv.VLP-SS-MUC1 showed a modest trend of delayed tumour growth and improved survival compared to the control group. However, these differences were not statistically significant. Once tumours reached 150 mm^2^, TILs were isolated from the tumour tissue of five mice/group and analysed by flow cytometry to examine T-cell exhaustion. Again, cells were gated on either CD3+ CD4+ cells or CD3+ CD8+ cells and the proportion of PD-1+, CD39+, LAG-3+, and TIM-3+ cells evaluated. No differences in T-cell exhaustion marker expression was observed between TILs isolated from the tumours of VLP-vaccinated and control mice. The remaining five tumours/group were subjected to CD3 IHC to assess T-cell infiltration within tumours. In VLP-vaccinated mice, a trend of increased CD3+ cell infiltration was observed within the total tumour, as well as the peritumoural and intratumoural areas (Figure 4e). However, these differences were not statistically significant when compared to VP60 vaccinated controls. Hence, while vaccination with surv.VLP-SS-MUC1 could induce protective anti-tumour immunity in WT mice, this VLP-based vaccine was ineffective as a monotherapy in MUC1 transgenic mice.

## 4. Discussion

The delivery of tumour antigens using virus-like particles (VLPs) has shown promise as an immunotherapy, inducing enhanced anti-tumour immune responses compared to the unformulated delivery of free peptides [29,30,31]. In this study, we hypothesized that the simultaneous delivery of two peptide antigens on VLPs could further enhance the anti-tumour efficacy of VLP-based vaccination by targeting a wider variety of tumour cell populations. The two TAAs mucin-1 (MUC1) and survivin were selected as targets for breast cancer immunotherapy due to their overexpression in approximately 90% of breast cancers [19,32]. The MUC1 peptide sequence was selected based on previously published reports [18,25,26,27,28,33] and included the aberrant glycosylation structure Tn (αGalNAc-Thr). Here, we describe the generation of VLPs co-delivering two tumour antigens (MUC1 and survivin) and evaluate their efficacy as an immunotherapy in a murine breast cancer model.

The delivery of free peptide antigens alone is often insufficient to break tolerance to an endogenous self antigen. However, peptide delivery via VLPs in combination with adjuvation has been shown to generate strong anti-tumour immune responses [21,34]. Initially, the recombinant insertion of two antigens into the RHDV VP60 capsid protein was considered, as we previously utilised this strategy in a VLP-based vaccine for colorectal cancer [21]. However, the αGalNAc-Thr glycosylation on the threonines of the MUC1 peptide is an important feature that specifically elicits T-cell reactivity against BC cells, as these aberrant sugars are absent from MUC1 expressed by healthy tissues. As VLPs were expressed in Sf insect cells, which are limited in their ability to express complex sugars compatible with mammalian cells [35], only the survivin epitope TVSEFLKL (survivin_97-104_) was recombinantly inserted into the VP60 capsid protein. The aberrantly glycosylated SAPDT(GalNAc)RPAPGST(GalNAc)APPA MUC1 peptide was subsequently conjugated onto the surface of assembled surv.VLP particles. For this conjugation, we utilised the reducible HYN-SS linker, which was previously shown by our group to cleave only at intracellular concentrations of GSH, while remaining stable at extracellular concentration [3]. This feature enables the co-delivery of both antigens into the same APC for processing and presentation to T cells, while enabling the preferential release of the MUC1 peptide inside the cell. Insertion of the survivin epitope as well as subsequent conjugation did not affect particle formation. However, there was increased clustering of the particles following conjugation. This clustering did not affect the uptake of VLPs by APCs (data not shown) and had no effect on the generation of antigen-specific T-cell responses in the subsequent cytotoxicity and tumour growth assays.

The establishment of targeted antigen-specific cytotoxicity by peptide vaccines is a reliable early indicator of their ability to elicit an anti-tumour immune response. We detected cytotoxicity against both the survivin and MUC1 peptides in all mice vaccinated with VLPs delivering either one or both peptides, as well as no off-target cytotoxicity against VP60, confirming the generation of antigen-specific cell-mediated immunity. The generation of cytotoxic CD8^+^ T-cell responses was also assessed in MUC1.Tg mice. These mice express the human MUC1 gene and represent an established model for testing whether MUC1 vaccines can break tolerance against endogenously expressed MUC1 protein [23]. Importantly, MUC1.Tg mice vaccinated with surv.VLP-SS-MUC1 demonstrated antigen-specific cytotoxicity against both survivin and MUC1 peptides. These data suggest that in a prophylactic vaccination setting, the endogenous expression of MUC1 does not impair the generation of targeted immunity against the MUC1 peptide. The absent or low cytotoxicity induced by vaccination with the free peptides confirmed that VLP-based delivery of peptide antigens enhances the generation of antigen-specific T-cell responses in vivo.

In the subsequent therapeutic tumour trial, we observed differences between VLPs delivering either one or both peptide antigens. As hypothesized, wild-type mice vaccinated with surv.VLP-SS-MUC1 showed delayed tumour growth and improved overall survival compared to mice vaccinated with VLPs delivering either survivin or MUC1 peptides alone. When comparing the expression of exhaustion markers in TILs isolated from surv.VLP-SS-MUC1 vaccinated mice with those isolated from mice vaccinated with empty VLPs (VP60 control), only minimal differences were observed. Clearer differences in T-cell exhaustion may have been detected if TILs had been analysed while tumour growth was still suppressed in the earlier stages of the trial, rather than when tumours had escaped immune control and had reached the end point of 150 mm^2^. Future studies will focus on elucidating the effects of VLP vaccination on TIL phenotype and function throughout the duration of tumour growth.

While the VLP vaccine delivering both antigens induced a significant survival benefit in wild-type mice, it was unable to suppress tumour growth in MUC1.Tg mice. Using the MUC1.Tg mice as a model demonstrated that administering the VLP vaccine alone—even though it co-delivered both survivin and MUC1 peptides—was insufficient to break tolerance towards endogenously expressed MUC1 protein. Similar to the trial in wild-type mice, no differences in the phenotype of TILs were detected when tumours were analysed at their 150 mm^2^ end point. These findings suggest that additional strategies are required to break tolerance when the tumour antigen(s) targeted by vaccination are endogenously expressed self-proteins (i.e., TAAs). One promising approach would be to combine VLP vaccination with immune checkpoint blockade agents, such as anti-PD-1, anti-PD-L1 or anti-CTLA-4 antibodies [36]. Indeed, this combinatorial approach has successfully enhanced anti-tumour immune responses against other endogenously expressed TAAs targeted in other vaccine formulations—both in preclinical and clinical studies [37,38,39,40]. Alternatively, targeting multiple mutated neoantigens (as opposed to TAAs) using VLP-based vaccines is another promising avenue for BC immunotherapy. Importantly, these tumour-specific antigens are highly immunogenic and recognised as ‘foreign’ by the immune system as reactive T cells are exempted from central tolerance [41].

In summary, this study showed that VLPs simultaneously delivering two cancer antigens induce a targeted anti-tumour immune response. Delivery of two antigens was superior in generating a therapeutically effective anti-tumour response against breast cancer compared to the single delivery of either antigen alone. Therefore, using VLPs as a platform for the co-delivery of multiple antigens represents a promising approach to improving BC immunotherapy, and has the potential to be used as an integral part of combination therapy in the future.

## Figures and Tables

**Figure 1 vaccines-09-00463-f001:**
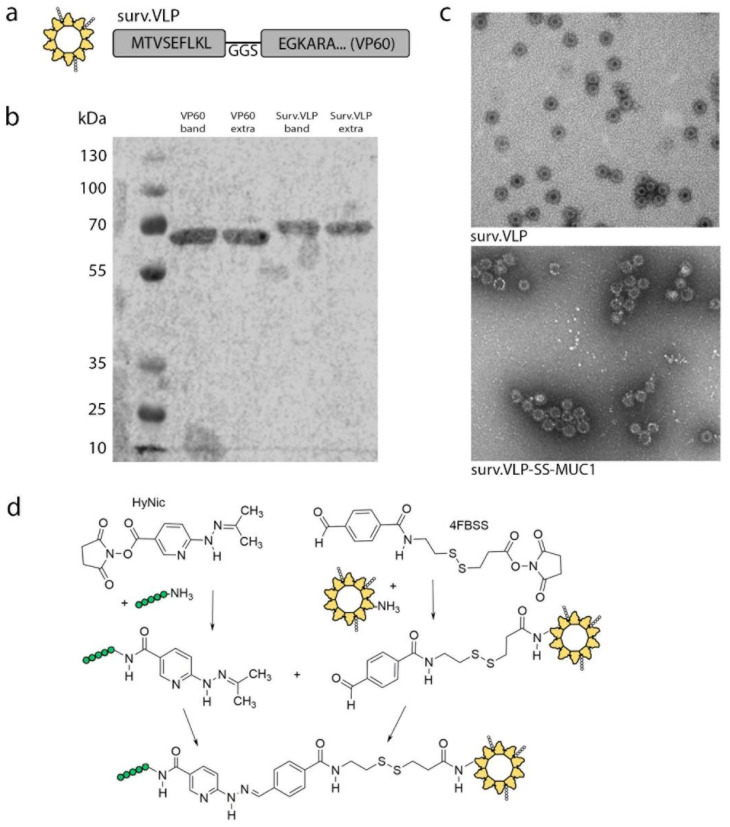
surv.VLP-SS-MUC1 design and expression. (**a**) Diagram outlining the structure and position of survivin for surv.VLP. (**b**) Western blot for surv.VLP and control VLP. (**c**) Electron microscopy to confirm particle formation of surv.VLP and surv.VLP-SS-MUC1. (**d**) Conjugation strategy to attach MUC1 peptide on surv.VLP via bis-aryl-hydrazone linker.

**Figure 2 vaccines-09-00463-f002:**
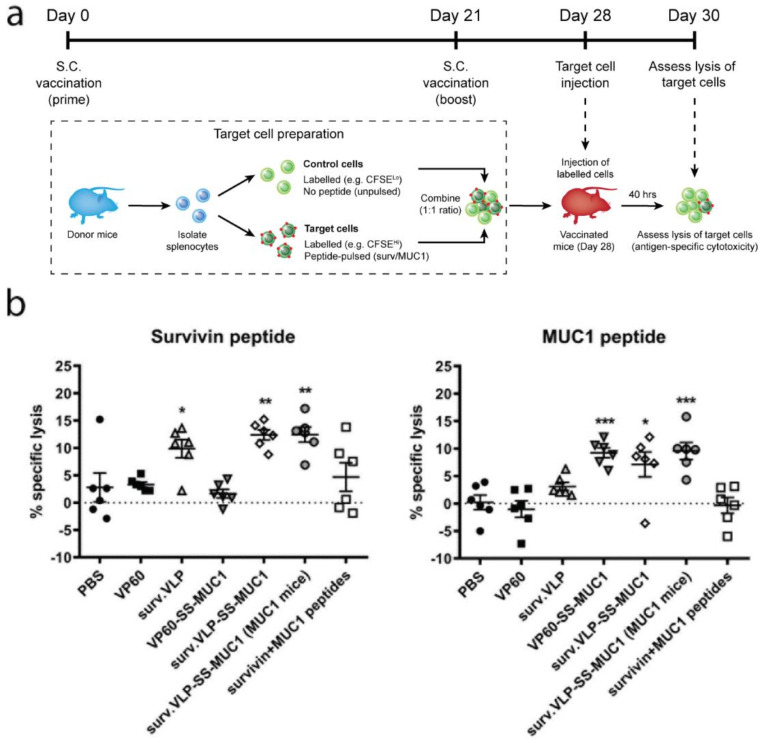
In vivo cytotoxicity induced by surv.VLP-SS-MUC1. (**a**) Timeline of cytotoxicity assay. On day 0, mice (*n* = 6 per treatment group) were vaccinated with VLPs delivering one or two antigens or empty VLP controls (VP60). 21 days later, mice received a boost with the same treatment and, on day 28, mice were injected with target cells, pulsed with either survivin or MUC1 peptide. Mice were culled 48 h after target cell injection, and splenocytes were analysed. (**b**) Specific lysis of target cells was calculated by comparing unpulsed to target cell ratio. Results show percent specific lysis of vaccinated treatment groups compared to the VP60 group ± SEM. Statistical significance was determined by one-way ANOVA with Tukey’s post hoc test, *** *p* < 0.001, ** *p* < 0.01, * *p* < 0.05.

**Figure 3 vaccines-09-00463-f003:**
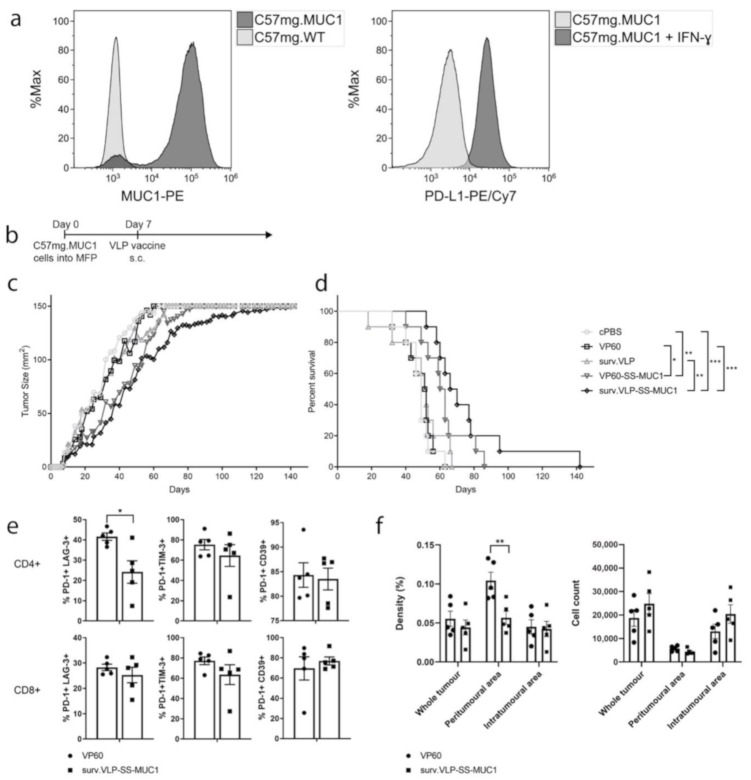
Anti-tumour response of surv.VLP-SS-MUC1 in therapeutic tumour trial. (**a**) Expression of MUC1 and PD-L1 (following IFN-y stimulation) by C57mg.MUC1 cells. (**b**) Timeline of tumour trial. C57/BL6 mice (*n* = 10 mice per treatment group) were inoculated with C57mg.MUC1 tumour cells in the mammary fat pad (MFP). Once tumours were palpable (day 7), mice were vaccinated with VLPs delivering one or two antigens or controls. Tumour growth (**c**) and overall survival (**d**) for each treatment group were recorded. Once tumour size reached 150 mm^2^ mice were euthanised and tumours analysed for TILs by flow cytometry (**e**) and IHC for CD3 (**f**). Statistical significance was determined by a log-rank (Mantel–Cox) test for survival and Mann–Whitney tests for TIL phenotype and IHC. *** *p* < 0.001, ** *p* < 0.01, * *p* < 0.05.

**Figure 4 vaccines-09-00463-f004:**
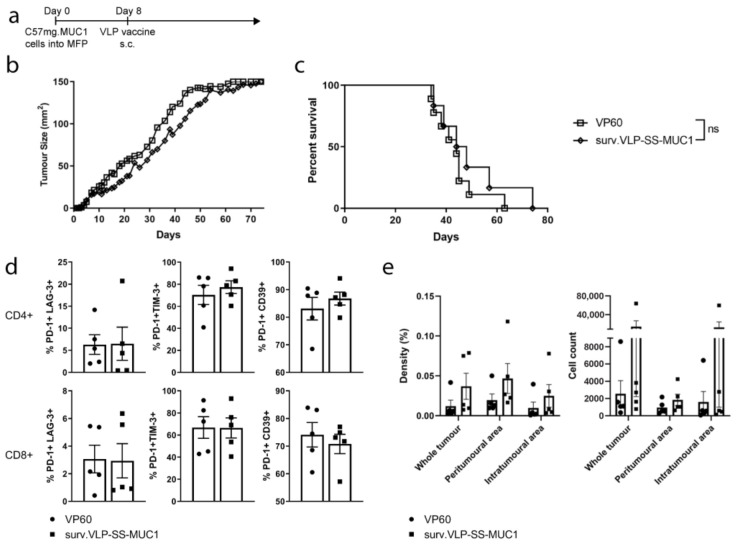
Anti-tumour responses in MUC1.Tg mice induced by surv.VLP-SS-MUC1 therapeutic vaccination. (**a**) Timeline of tumour trial. MUC1.Tg mice (*n* = 10 mice per treatment group) were inoculated with C57mg.MUC1 tumour cells in the mammary fat pad (MFP). Once tumours were palpable (day 8), mice were vaccinated with surv.VLP-SS-MUC1 or empty VLPs as a negative control (VP60). Tumour growth (**b**) and overall survival (**c**) for each treatment group were recorded. Once tumours reached a size of 150 mm^2^, mice were euthanised and tumours analysed for TILs by flow cytometry (**d**) and IHC for CD3 (**e**). Statistical significance was determined by a log-rank (Mantel–Cox) test for survival and Mann–Whitney tests for TIL phenotype and IHC.

## Data Availability

The data presented in this study are available in the manuscript and associated supplementary materials.

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
