# Peer review of "Delivering Two Tumour Antigens Survivin and Mucin-1 on Virus-Like Particles Enhances Anti-Tumour Immune Responses"

_vaccines, 2021, doi:10.3390/vaccines9050463_

Round 1

Reviewer 1 Report

In this manuscript the authors designed a protocol based on a delivery platform based on a virus-like particles (VLP) conjugated to an aberrantly glycosylated mucin  (MUC1) and surviving (a recombinantly expressed murine), composing the complex VLP-SS-MUC1. Thus, a VLP-based BC vaccine was developed, which could be able to develop a therapeutically effective anti-tumour response against breast cancer.

The obtained data shows that the designed protocol reached effective results concerning the delay of progress of breast cancer and survival of experimental animals. Based on this, the authors indicate that the results point to a potential used of VLP-based BC vaccine as part of combination therapy for cancer breast.

Just one recommendation: To reduce repetition of methodology in the results section.

Author Response

In this manuscript the authors designed a protocol based on a delivery platform based on a virus-like particles (VLP) conjugated to an aberrantly glycosylated mucin (MUC1) and survivin (a recombinantly expressed murine), composing the complex VLP-SS-MUC1. Thus, a VLP-based BC vaccine was developed, which could be able to develop a therapeutically effective anti-tumour response against breast cancer.

The obtained data shows that the designed protocol reached effective results concerning the delay of progress of breast cancer and survival of experimental animals. Based on this, the authors indicate that the results point to a potential used of VLP-based BC vaccine as part of combination therapy for cancer breast.

Just one recommendation: To reduce repetition of methodology in the results section.

Thank you for this recommendation, we have reduced the repetition of methodology in our results section as advised (refer to Tracked Changes in revised manuscript document). 

Reviewer 2 Report

The authors investigated delivery of two tumor antigens: surviving and MUC1 on virus-like particles (VLPs) to enhance anti-tumor immune response. They used previously developed VLPs and performed recombinant insertion of survivin epitope and attached aberrantly glycosylated MUC1. It is an interesting approach and the manuscript is well prepared. Though, I still have some questions:

  • The authors used ultracentrifugation for VLP purification, while it seems that HPLC would be a better method for that. Is there any specific reason for the choice of technically more demanding and slower ultracentrifugation?
  • The another surprise is using mass spectrometry for detection of survivin epitope. Here, immunological methods would be a better fit for such detection.
  • Description of in vivo toxicity is unclear. I would suggest to add a graphical presentation of this experiment.
  • Why C57mg.WT cells were not transplanted to MUC1.Tg mice?
  • What is a reason for a selection of 100μg dos of VLPs? Why it was not dose escalation study? Might be this dose is too little to achieve positive therapeutic effect?
  • It was missed checking reactivity of TIL to survivin and MUC1, although the authors are planning to include it in future studies.
  • Was there visible antigen escape in tumor after vaccine administration, which would be an explanation for a poor activity of investigated VLPs.
  • Is there advantage of having two antigens on one VLP in comparison to have two different VLPs? Is there any upper limit of VLP dose?
  • What is an expected duration of immune response induced by VLPs?
  • The authors stated that the addition of checkpoint inhibitors was able to break tolerance of VLP to autoantigen. But there is missing control with checkpoint inhibitors only to exclude that the therapeutic effect is only related to checkpoint inhibitors and not to VLPs.

Author Response

The authors investigated delivery of two tumor antigens: survivin and MUC1 on virus-like particles (VLPs) to enhance anti-tumor immune response. They used previously developed VLPs and performed recombinant insertion of survivin epitope and attached aberrantly glycosylated MUC1. It is an interesting approach and the manuscript is well prepared. Though, I still have some questions:

  • The authors used ultracentrifugation for VLP purification, while it seems that HPLC would be a better method for that. Is there any specific reason for the choice of technically more demanding and slower ultracentrifugation?

Thank you for raising this point. We have previously used both HPLC and FPLC to purify our VLPs, however these methods gave much lower yields of VLP than centrifugation, which is why we have retained a centrifugation purification protocol.

  • Another surprise is using mass spectrometry for detection of survivin epitope. Here, immunological methods would be a better fit for such detection.

We believe that this approach was appropriate, given that mass spectrometry is widely regarded as gold-standard methodology for the detection of peptide epitopes. However, we agree that detection using immunological methods would also be appropriate. In this regard, our results for the in vivo cytotoxicity assay (Fig. 2B) also confirm the presence of the survivin epitope in our VLPs ‘immunologically’, as these could elicit antigen-specific cytotoxicity against survivin.

  • Description ofin vivo toxicity is unclear. I would suggest to add a graphical presentation of this experiment.

Thank you for this suggestion. We have added a graphical representation to clarify this experiment (refer to Figure 2A).

  • Why C57mg.WT cells were not transplanted to MUC1.Tg mice?

In this experiment, we wanted to assess the efficacy of surv.VLP-SS-MUC1 when human MUC1 was expressed as a “self” antigen (i.e. subjected to tolerance mechanisms). For this reason, we only implanted C57mg cells expressing human MUC1 (C57mg.MUC1) into these mice. However, in future studies we will include implanting C57mg.WT cells as a control, enabling us to determine the relative contribution of anti-survivin versus anti-MUC1 immune responses to tumour control (C57mg.WT would show the contribution of anti-survivin responses only).

We have additionally added our Supplementary Figures 2 and 3, which show titration of C57mg.WT and C57mg.MUC1 cells in C57BL/6 mice, as well as the therapeutic tumour trial in C57BL/6 mice using C57mg.WT tumour cells as specified in the paper text. We apologise for the oversight of not including these when we first submitted the manuscript.

  • What is a reason for a selection of 100μg dose of VLPs? Why it was not dose escalation study? Might be this dose is too little to achieve positive therapeutic effect?

Our group has carried out extensive research on RHDV VLP-based vaccines and we have previously performed dose escalation and prime-boost studies. Our previous work has shown that 50ug or higher produces a robust response in a range of in vivo assays, on which the dose of 100 ug used in this study was based. We performed these studies using a variety of antigens (gp100, survivin, topoisomerase, SIINFEKL) and are therefore confident of the 100ug dose. Please also see the references 9, 21, 22, 29-31 and 34 that we included in the manuscript.

  • It was missed checking reactivity of TIL to survivin and MUC1, although the authors are planning to include it in future studies.

We agree that this data would have been informative, and we plan to perform these studies in the future.

  • Was there visible antigen escape in tumor after vaccine administration, which would be an explanation for a poor activity of investigated VLPs.

We did not assess the possibility of antigen escape variants emerging in response to VLP vaccination. However, this is certainly a possibility and we agree that this may explain their limited therapeutic efficacy.

  • Is there advantage of having two antigens on one VLP in comparison to have two different VLPs? Is there any upper limit of VLP dose?

Delivery of both antigens on one VLP has the advantage that both antigens are taken up in equal proportions by APCs. This reduces the chance of uptake of one VLP being favoured by APCs, which could potentially result in a ‘biased’ immune response against only one tumour antigen. Additionally, from a practical point of view, VLP only need to be produced and purified once, making it more translatable into the clinic and desirable from a manufacturing perspective.

  • What is an expected duration of immune response induced by VLPs?

Thank you for raising this point. So far, we have not performed any long-term studies to assess the duration of VLP-induced immune responses against survivin and MUC1. In our previous studies, tumour control was shown to persist long-term (200 days post VLP injection; Donaldson et al., 2017), but we agree that long-term data would be informative in this particular study. We plan to perform experiments to address this aspect in the future.

  • The authors stated that the addition of checkpoint inhibitors was able to break tolerance of VLP to autoantigen. But there is missing control with checkpoint inhibitors only to exclude that the therapeutic effect is only related to checkpoint inhibitors and not to VLPs.

Thank you for raising this point. We did not add a group of checkpoint inhibitor only to this study, as our approach was to just look at enhancing the VLP vaccine. Addition of another group was also in conflict with the 3Rs of animal research (Replacement, Reduction and Refinement), and we endeavoured to keep the use of animals to a minimum. However, addition of a checkpoint inhibitor only group would have been informative, and we plan to include it in future studies.

Round 2

Reviewer 2 Report

The authors properly addressed all my points but the last one. The advantage of checkpoint blockade in comparison to the author’s vaccine results is so huge that it is only a source of confusion, that checkpoint blockade is additionally helpful, ad in fact checkpoint blockade may completely override the vaccine effect. Therefore, the data with vaccine and checkpoint blockade without the checkpoint blockade are incomplete. So, I recommend to add a group with checkpoint blockade only, or to remove this experiment from the manuscript, and to include in one of the future manuscripts, when the appropriate control is included. Even, without this experiment, the manuscript includes a lot of valuable data of interest for readers.

Author Response

Comments and Suggestions for Authors

The authors properly addressed all my points but the last one. The advantage of checkpoint blockade in comparison to the author’s vaccine results is so huge that it is only a source of confusion, that checkpoint blockade is additionally helpful, ad in fact checkpoint blockade may completely override the vaccine effect. Therefore, the data with vaccine and checkpoint blockade without the checkpoint blockade are incomplete. So, I recommend to add a group with checkpoint blockade only, or to remove this experiment from the manuscript, and to include in one of the future manuscripts, when the appropriate control is included. Even, without this experiment, the manuscript includes a lot of valuable data of interest for readers.

We agree with the Reviewer’s concerns and accept that the final experiment in our manuscript is incomplete without a checkpoint blockade only control (anti-PD-L1 only). Although we would like to repeat this experiment to include a checkpoint blockade only group for this manuscript, we have very recently relocated our lab from New Zealand to Australia while the lead author of this manuscript has moved to Germany. Consequently, we are unable to repeat this experiment within a suitable timeframe. However, we believe that the manuscript still contains valuable data of interest regarding the design and synthesis of our VLP co-delivering two tumour antigens. Furthermore, the failure of our VLP to induce effective anti-tumour immunity as a monotherapy in the MUC1.Tg mouse model (i.e. to break tolerance when targeted antigens were expressed as endogenous self-proteins) has important implications for the therapeutic targeting of tumour-associated antigens (particularly overexpressed self-proteins) with vaccine formulations.

For these reasons, we have opted to remove the final experiment from the manuscript (and all references in the manuscript to this experiment) as suggested. We have also revised our discussion to reflect our main findings and their implications without this experiment. We would also like to thank Reviewer 2 for their constructive criticism, and we plan to definitively address the potential benefit of combining VLP-based vaccination with immune checkpoint blockade in our future work.

Submission Date

29 March 2021

Date of this review

25 Apr 2021 04:38:19

Round 3

Reviewer 2 Report

No further comments